# Comparative Study on the Effects of Various Modified Admixtures on the Mechanical Properties of Styrene-Acrylic Emulsion-Based Cement Composite Materials

**DOI:** 10.3390/ma13010008

**Published:** 2019-12-18

**Authors:** Tengjiao Wang, Jinyu Xu, Congjin Zhu, Weibo Ren

**Affiliations:** 1School of Aeronautical Engineering, Air Force Engineering University, Xi’an 710038, China; 2College of Mechanics and Civil Architecture, Northwest Polytechnic University, Xi’an 710072, China

**Keywords:** styrene-acrylic emulsion, cement composite, airport joint sealant, admixture, bond property, tensile property, shear performance

## Abstract

This study carried out tensile tests at definite elongation, tensile and shear tests on 4 admixture-modified styrene-acrylic emulsion-based cement composites (SECCs), and measured the strength, deformation, and energy consumption indexes of test specimens, so as to investigate the influences of coalescing agent, plasticizer, silane coupling agent, and nanometer aluminium oxide on the bond, tensile, and shear mechanical properties of the test specimens. Additionally, the Field Emission Scanning Electron Microscope (FE-SEM) test and Mercury Intrusion Porosimetry (MIP) test were conducted on the composite material specimens, to analyze the microscopic mechanism of different admixtures in modifying the mechanical properties of the SECC. The results suggested that the addition of coalescing agent, plasticizer, silane coupling agent, and nanometer aluminium oxide improved the bond, tensile and shear properties of the SECC specimens to various degrees. Of them, the coalescing agent promoted the mutual cross-linking of organic polymers with inorganic products, and optimized the transition interface to enhance the comprehensive mechanical properties of the test specimens; by contrast, nanometer aluminium oxide developed secondary hydration reaction with the inorganic products, and refined the pore structure to modify the mechanical properties of test specimens. Therefore, both of them achieved significant modification effects. Typically, the optimal bond properties of FFAMC, PLMC, SCAMC, and NAMC test specimens were achieved at the coalescing agent, plasticizer, silane coupling agent, and nanometer aluminium oxide addition amounts of 4%, 1.5%, 3%, and 1%, respectively. Besides, the improving effects of different admixtures on the tensile property of SECC specimens followed the order of coalescing agent > nanometer aluminium oxide > plasticizer > silane coupling agent, with the optimal addition amounts of 4%, 1.5%, 1%, and 2%, respectively. In addition, the improving effects of different admixtures on the shear performance of SECC specimens followed the order of coalescing agent > nanometer aluminium oxide > silane coupling agent > plasticizer, with the optimal addition amounts of 4%, 1.5%, 1%, and 1%, respectively.

## 1. Introduction

Cement concrete pavement slab joint is the weak link in airport pavement [1,2,3], and the pavement slab is usually subjected to damages such as peeling, fragmentation, pumping, and faulting due to water seepage, fracture, and aging of calking [4,5,6,7]. And the damaged airport pavement is the mainly threat when taking off and landing an aircraft. To solve the water seepage and fracture problem of joint sealant, domestic and foreign researchers have carried out plenty of experimental studies. At present, the frequently used airport pavement joint sealants [8,9,10] include silicone [11,12], polysulfide [13,14], polyurethane [15,16], and poly-thionocarbamate [17,18] organic polymer materials. These materials possess favorable performance in impermeability and deformation, but they usually have poor strength and weather resistance [19]. Therefore, it is of great application prospect to develop a novel composite material with high strength, deformation performance, and water resistance.

Polymer emulsion is characterized by the properties of rapid film formation, high film strength, great toughness, and good durability [20,21,22]. When it is mixed with cement, the organic polymers develop mutual cross-linking reaction with the inorganic molecules [23,24,25] to produce the polymer-based cement composite material, which possesses the deformation energy adsorption property of organic polymer and the high strength durability of inorganic silicate material, and can thereby serve as a novel joint sealant [26,27]. The styrene-acrylic emulsion-based cement composite (SECC) prepared with styrene-acrylic emulsion as the organic matrix, while cement as the enhancer, has favorable bond property and flexibility, high strength, and superb durability [28,29,30], which is thereby used to manufacture the airport pavement joint sealant. However, the styrene-acrylic emulsion is soft with few active groups [31,32,33], which has less reaction points with cement; as a result, it is difficult to sufficiently cement and cure the styrene-acrylic emulsion and the cement particles, and a large amount of pores are likely to be produced at the interface of polymer film and inorganic product, thus affecting the mechanical properties of the SECC. Consequently, it is necessary to add other admixtures to modify and optimize the SECC.

Coalescing agent improves the mobility and deformation performance of organic polymer, reduces the intermolecular repulsion, and enhances the film-forming cementation between organic macromolecule and the inorganic molecule [34,35]. Plasticizer improves the polymer plasticity and flexibility, which weakens the secondary physical bond energy between the organic polymers to reduce the intermolecular force and increase the contact area between organic molecule and inorganic molecule [36,37]. Silane coupling agent is a kind of organic silicide with organic functional group and siloxy, which is able to bridge the organic phase and inorganic phase, optimize the transition interface between two phases, and enhance the compatibility and integrity [38], due to the reactivity between organic functional group and organic matter, as well as between siloxy and inorganic molecule. Nanometer aluminium oxide can absorb and bond the organic molecule with inorganic molecule, improve the transition interface between organic and inorganic phase, and promote the reaction coalescence between organic polymer and inorganic matter [39]. At present, domestic and foreign scholars have carried out research on SECC, but there are a few studies on how to improve its properties, even there are almost no reports of comparative study on the effects of various modified admixtures. On this basis, this paper treated strength, deformation, and energy consumption indexes as the research indexes to carry out tensile tests at definite elongation, tensile and shear test, so as to comparatively investigate the effects of coalescing agent, plasticizer, silane coupling agent, and nanometer aluminium oxide on improving the bond, tensile and shear properties of the SECCs, considering that joint sealant mainly undertook the tensile and shear loads. In addition, the optimal addition amounts of various admixtures were also determined, and the microscopic mechanism of different admixtures in modifying the mechanical properties of SECCs was also discussed based on Field Emission Scanning Electron Microscope (FE-SEM) test and Mercury Intrusion Porosimetry (MIP) test.

## 2. Tests

### 2.1. Test Materials

The clean tap water in the laboratory was used in the tests; the Ba River medium sands with the fineness modulus of 2.8 were used as the fine aggregates, with the silt content of 1.3%. The limestone gravels with the particle composition of 14–22 cm were used as the coarse aggregates. The cement adopted the ordinary Portland cement at grade 42.5, with the density of 3.1 g/cm^3^. The SN-DISPERSANT 5040 type dispersing agent and the NOPCO NXZ type defoamer (San Nopco Ltd., Tokyo, Japan) were also utilized in tests. The Acronal S400F ap-type styrene-acrylic emulsion (BASF Corporation, Berlin, Germany) was used as the polymer emulsion, and the related technical indexes are shown in Table 1. The admixtures used were alcohol ester-12 (DN-12) coalescing agent, dioctyl phthalate (DOP) plasticizer, anilmomethyl triethoxy silane (ND-42) silane coupling agent, and VK-L100G-type nanometer aluminium oxide. The major technical indexes are shown in Table 2, Table 3, Table 4 and Table 5.

### 2.2. Preparation of Test Specimens 

The above mentioned materials were used in the tests. The modified SECCs with different admixtures were prepared according to the mixture ratios in Table 6. The precise preparation process was shown below.

The styrene-acrylic emulsion and dispersing agent were added into the blender at the same time, to stir for 2 min, and the evenly dispersed styrene-acrylic dispersing solution was obtained. Subsequently, the defoamer was added, to stir for 1 min, and mixed sufficiently. Finally, the coalescing agent, plasticizer, and silane coupling agent were added, respectively, to stir for 5 min, so as to obtain the mixed solution with different admixtures.

The cement was mixed with talcum powder and stirred evenly, subsequently, the mixture was added into the prepared mixed solution to stir for 5 min at low speed and for 10 min at high speed, so as to obtain the evenly mixed modified SECC with stable properties, such as film-forming auxiliary modified styrene-acrylic emulsion-based cement composite (FFAMC), plasticizer modified styrene-acrylic emulsion-based cement composite (PLMC), silane coupling agent modified styrene-acrylic emulsion-based cement composite (SCAMC), nanometer aluminium oxide-modified styrene-acrylic emulsion-based cement composite (NAMC). Of them, no modifying admixture was added into the liquid when preparing the NAMC, instead, the nanometer aluminium oxide particles were mixed with cement and talcum powder, and the remaining steps were the same as those above.

The prepared modified SECC material was poured into the mold cavity and cured for 28 days at standard conditions (temperature of 21–25 °C, relative humidity of 45–55%). The composite material test specimen used in the tests was obtained after form removal, and the prefabricated cement mortar substrate was bonded on both sides of the test specimen, as shown in Figure 1.

### 2.3. Test Methods 

Tensile at definite elongation test: the prepared test specimen was placed into the tensile test mold with fixed tensile displacement, shown in Figure 2, and stretched to 60% deformation quantity of the original width at the rate of 5 mm/min. Then, the location block was inserted, followed by 24 h standing under standard conditions. Later, the test specimen was examined for damage, the location block was removed in the absence of test specimen damage, then, the test specimen width after recovering the elasticity was measured, and the elasticity recovery rate of test specimen was calculated according to formula (1). Each value represented the average value from three independent tests.
(1)$$R_{e}=\frac{w_{1}-w_{2}}{w_{1}-w_{0}}\times 100\%,$$
here $w_{0}$, $w_{1}$ and $w_{2}$ stand for the initial width, width at definite elongation, and width after elasticity recovery of the test specimen, respectively.

Tensile test: the HS-3001B electronic tensile device was adopted for the tensile test of the test specimen in accordance with the national standard, as shown in Figure 3 and Figure 4.

Shear test: the HS-3001B electronic shear device was adopted for the shear test of the test specimen in accordance with the national standard, as shown in Figure 5. 

Microscopic test: the COX I EM-30 scanning electron microscope (SEM, CUSEM company, South Korea) was used for SEM tests of the SECC material test specimens modified by various admixtures, so as to analyze the microstructures of the test specimens. Before the test, the ETD-800 automatic ion sputtering device (Woshide Technology Co. Ltd., Beijing, China) was used to perform a gold spray treatment for 90 s on the newly cut sample to improve the conductivity of the surface of the samples, and then the electron microscope was used to observe the samples with different magnification degree. Meanwhile, the Pore Master-33 mercury injection (Conta Instruments, the United States) apparatus was adopted for the mercury injection tests for the SECC material test specimens modified by various admixtures, so as to analyze the pore structures of the test specimens. Before the test, the samples were weighed with a precision electronic balance, and then they were placed in a low-pressure station and a high-pressure station for mercury pressure test analysis. The test pressure ranged from 20 to 30,000 psi and the mercury contact angle was 140°.

## 3. Test results and analysis 

### 3.1. Bond Property 

Figure 6 shows the tensile at definite elongation bond failure morphologies of the SECC material test specimens modified by different admixtures. It was observed from the figure that, the test specimen had good bond property at the low addition amount of coalescing agent, but the test specimen was subjected to mild cohesive failure and bond failure at the great addition amount, such as FFAMC-4. In addition, the addition amounts of plasticizer and silane coupling agent had marked effect on the bond property of the test specimens, the test specimen had good bond performance at low addition amount, but both test specimens were subjected to severe bond failure at the great addition amount, such as PLMC-4 and SCAMC-4. Additionally, the test specimen had favorable bonding property at the addition amount of nanometer aluminium oxide of <2%, but obvious bond failure begun to appear at the addition amount of >2%, like NAMC-5. Based on the influence rule of different admixtures on the elasticity recovery rate of test specimen, as shown in Figure 7, the elasticity recovery rates of test specimens FFAMC and PLMC were increasing, among which, the elasticity recovery rate of PLMC test specimen showed almost linear increase at the plasticizer addition amount of <3%, and that remained almost unchanged at the addition amount of >3%. The elasticity recovery rate for SCAMC test specimen was decreased with the increase in silane coupling agent addition amount, that was markedly reduced to less than 60% at the silane coupling agent addition amount of >1%, which did not satisfy the requirement of elasticity recovery rate of SECC. With the increase in nanometer aluminium oxide addition amount, the elasticity recovery rate of NAMC test specimen showed a first increasing and then decreasing trend, the maximal elasticity recovery rate was achieved at the addition amount of 1.5%, but that was less than 60% when the addition amount was increased to 2.5%, which did not satisfy related requirement. In summary, the optimal bond properties of FFAMC, PLMC, SCAMC, and NAMC test specimens were the optimal at the coalescing agent, plasticizer, silane coupling agent, and nanometer aluminium oxide addition amounts of 4%, 3%, 1% and 1.5%, respectively.

### 3.2. Tensile Property 

The material tensile property was characterized from the aspects of strength, deformation and energy consumption using three indexes, namely, tensile strength, tensile peak strain and tensile toughness. The tensile strength is the peak stress achieved during the tensile process of test specimen, the tensile peak strain is the strain value of the test specimen under the tensile peak stress, and the tensile toughness represents the area enclosed by the tensile stress strain curve of the test specimen and the horizontal axis. Figure 8 displays the influence rule of different admixtures on the tensile strengths of SECC test specimens. As observed, with the increases in coalescing agent, silane coupling agent and nanometer aluminium oxide addition amounts, the tensile strengths of FFAMC, SCAMC and NAMC test specimens were increasing. The tensile strengths of FFAMC and SCAMC test specimens were apparently increased when the coalescing agent silane coupling agent addition amounts were >2%; when the addition amount was increased to 4%, compared with SECC, the tensile strengths of FFAMC and SCAMC test specimens were increased by 28.3% and 62.2%, respectively. When the nanometer aluminium oxide addition amount was >0.5%, the tensile strength of NAMC test specimen was evidently increased; when the addition amount was increased to 2.5%, compared with SECC, the tensile strength of NAMC test specimen was elevated by 6.4%. The tensile strength of PLMC test specimen was markedly decreased with the increase in plasticizer addition amount, besides, the higher plasticizer addition amount led to a greater decreasing amplitude, and the tensile strength was decreased by 29.8% at the plasticizer addition amount of 4%. Thus, it was clear that, coalescing agent, silane coupling agent and nanometer aluminium oxide enhanced the tensile strength of SECC test specimens, and more obvious effects were obtained at the higher addition amounts; among which, silane coupling agent had the most obvious modification effect. In addition, plasticizer decreased the tensile strength of composite material test specimens, and a higher addition amount resulted in the more significant decreasing amplitude.

Figure 9 presents the influence rule of different admixtures on the tensile peak strains of SECC test specimens. Clearly, the tensile peak strains of FFAMC, PLMC, and NAMC test specimens showed a first increasing and then decreasing trend with the increases in coalescing agent, plasticizer and nanometer aluminium oxide addition amounts. Typically, the tensile peak strains of FFAMC, PLMC and NAMC test specimens peaked at the coalescing agent, plasticizer and nanometer aluminium oxide addition amounts of 3%, 1% and 1.5%, respectively; which were increased by 17.6%, 2.6% and 10.8% compared with SECC. When the coalescing agent addition amount was increased to 4%, the tensile peak strain of FFAMC test specimen was still greater than that of SECC; when the plasticizer and nanometer aluminium oxide addition amounts were increased to 4% and 2.5%, respectively, the tensile peak strains of PLMC and NAMC test specimens were markedly decreased; compared with SECC, they were reduced by 38.0% and 19.6%, respectively. The tensile peak strain of SCAMC test specimen showed a decreasing trend on the whole with the increase in silane coupling agent addition amount; that was the minimal at the silane coupling agent addition amount of 3%, which was reduced by 66.6% compared with that of SECC; while that was slightly increased when the addition amount was increased to 4%, but that was still smaller than the SECC test specimen. Thus, the addition of appropriate amounts of coalescing agent, plasticizer and nanometer aluminium oxide improved the tensile deformation property of the SECC material test specimens, among which, coalescing agent had the most obvious improving effect. In addition, the optimal tensile deformation properties of the FFAMC, PLMC and NAMC test specimens were attained at the coalescing agent, plasticizer and nanometer aluminium oxide addition amounts of 3%, 1%, and 1.5%, respectively, however, the addition of silane coupling agent reduced the tensile deformation property of test specimen.

Figure 10 displays the influence rule of different admixtures on the tensile toughness of SECC test specimens. It was observed from the figure that, the tensile toughness of FFAMC and NAMC test specimens showed a first increasing and then decreasing trend with the increases in coalescing agent and nanometer aluminium oxide addition amounts. Typically, the maximal tensile toughness of FFAMC and NAMC test specimens was achieved at the coalescing agent and nanometer aluminium oxide addition amounts of 3% and 1.5%, respectively, which was improved by 18.6% and 15.8% compared with the SECC test specimen, respectively. At the plasticizer addition amount of 1%, the tensile toughness of PLMC test specimen remained almost unchanged, but that was markedly decreased with the increase in plasticizer addition amount, and a greater addition amount led to a greater decreasing amplitude. When the plasticizer addition amount was increased to 4%, the tensile toughness of PLMC test specimen was reduced by 44.6% compared with that of SECC test specimen. The tensile toughness of SCAMC test specimen was greatly discrete with the increase in silane coupling agent addition amount, but that was apparently reduced relative to that of SECC test specimen. Thus, the addition of appropriate amounts of coalescing agent and nanometer aluminium oxide improved the tensile energy consumption properties of SECC test specimens; among which, the optimal tensile energy consumption properties of FFAMC and NAMC test specimens were achieved at the coalescing agent and nanometer aluminium oxide addition amounts of 3% and 1.5%, respectively. Meanwhile, PLMC test specimen had good property in tensile energy consumption at the plasticizer addition amount of <1%, but the addition of silane coupling agent reduced the tensile energy consumption property.

To sum up, the improving effects of various admixtures on the tensile property of SECC test specimens followed the order of coalescing agent > nanometer aluminium oxide > plasticizer > silane coupling agent. Typically, the optimal addition amounts of coalescing agent, nanometer aluminium oxide, plasticizer and silane coupling agent were 4%, 1.5%, 1% and 2%, respectively.

### 3.3. Shear Performance 

The material shear property was characterized from the aspects of strength, deformation and energy consumption using three indexes, namely, shear strength, shear peak strain and shear toughness. The shear strength is the peak stress achieved during the shear process of test specimen, the shear peak strain is the strain value of the test specimen under the shear peak stress, and the shear toughness represents the area enclosed by the shear stress strain curve of the test specimen and the horizontal axis. Figure 11 exhibits the influence rule of different admixtures on the shear strengths of SECC test specimens. It was observed that, with the increases in coalescing agent, silane coupling agent and nanometer aluminium oxide addition amounts, the shear strengths of FFAMC, SCAMC and NAMC test specimens were increasing; besides, the shear strength increasing amplitude was elevated as the addition amounts increased. The shear strengths of FFAMC, SCAMC and NAMC test specimens were elevated by 32.1%, 41.0% and 8.4%, respectively, compared with the SECC test specimen, at the coalescing agent, silane coupling agent and nanometer aluminium oxide addition amounts of 4%, 4% and 2.5%, respectively. After adding plasticizer, the shear strength of PLMC test specimen gradually was decreasing, and more obvious decrease was observed at a greater addition amount; the shear strength of PLMC test specimen was reduced by 37.3% at the plasticizer addition amount of 4%. Thus, it was clear that, coalescing agent, silane coupling agent and nanometer aluminium oxide enhanced the shear strength of SECC test specimens, and more obvious effects were obtained at the higher addition amounts; among which, silane coupling agent had the most obvious modification effect. In addition, plasticizer decreased the shear strength of composite material test specimens, and a higher addition amount resulted in the more significant decreasing amplitude.

Figure 12 displays the influence rule of different admixtures on the shear peak strains of SECC test specimens. Clearly, the shear peak strain of FFAMC test specimen showed an increasing trend with the increase in coalescing agent addition amount. Typically, the greatest shear peak strain was attained at the coalescing agent addition amount of 4%; which was increased by 35.6% compared with SECC test specimen. With the increase in nanometer aluminium oxide addition amount, the shear peak strain of NAMC test specimen showed a first increasing and then decreasing trend, and that peaked at the nanometer aluminium oxide addition amount of 1.5%, which was improved by 10.2% compared with the SECC test specimen. The shear peak strain was apparently decreased when the addition amount was increased to 2.5%, and that was smaller than the SECC test specimen. With the increases in plasticizer and silane coupling agent addition amounts, the shear peak strains of PLMC and SCAMC test specimens were decreasing at a large amplitude; the minimal shear peak strains of PLMC and SCAMC test specimens were obtained at the addition amounts of 4%, which were reduced by 47.3% and 37.9%, respectively, compared with SECC test specimen. Thus, the addition of appropriate amounts of coalescing agent, and nanometer aluminium oxide improved the shear deformation property of the SECC test specimens, among which, coalescing agent had the most obvious improving effect, and the more obvious effects were observed at the higher addition amount. In addition, the optimal addition amount of nanometer aluminium oxide was 1.5%, however, the addition of plasticizer and silane coupling agent reduced the shear deformation property of test specimen.

Figure 13 illustrates the influence rule of different admixtures on the shear toughness of SECC test specimens. It was observed from the figure that, the shear toughness of FFAMC test specimen was increasing with the increase in coalescing agent addition amount, and the best shear toughness was attained at the addition amount of 4%, which was improved by 47.0% compared with SECC test specimen. The shear toughness of NAMC test specimen showed a first increasing and then decreasing trend with the increase in nanometer aluminium oxide addition amount, and the greatest shear toughness was attained at the addition amount of 1.5%, which was improved by 3.4% compared with SECC test specimen; however, the shear toughness was markedly decreased when the addition amount was increased to 2.5%, and it was lower than SECC test specimen. With the increases in plasticizer and silane coupling agent addition amounts, the shear toughness of PLMC and SCAMC test specimens were gradually decreasing; compared with SECC test specimen, the shear toughness of PLMC and SCAMC test specimens was reduced by 64.7% and 18%, respectively, at the addition amounts of 4%. Clearly, the addition of appropriate amounts of coalescing agent and nanometer aluminium oxide improved the shear energy consumption properties of SECC test specimens; among which, coalescing agent had the most obvious improving effect, and the more obvious effects were observed at the higher addition amount. In addition, the optimal addition amount of nanometer aluminium oxide was 1.5%, however, the addition of plasticizer and silane coupling agent reduced the shear energy consumption property of test specimen.

In conclusion, the improving effects of various admixtures on the shear property of SECC test specimens followed the order of coalescing agent > nanometer aluminium oxide > silane coupling agent > plasticizer. Typically, the optimal addition amounts of coalescing agent, nanometer aluminium oxide, silane coupling agent and plasticizer were 4%, 1.5%, 1% and 1%, respectively.

## 4. Micro-Analysis

### 4.1. Microstructure 

Coalescing agent and nanometer aluminium oxide had better improving effects on the mechanical properties of the SECC material test specimens. Therefore, this study took FFAMC and NAMC test specimens as the examples to observe the microstructures of FFAMC and NAMC test specimens, and to analyze the microscopic mechanisms of coalescing agent and nanometer aluminium oxide in modifying the SECC test specimens. The microstructures of FFAMC and NAMC test specimens were shown in Figure 14. It was shown in Figure 14a that, after adding coalescing agent, the organic polymer film formed inside the test specimen cross-linked with the cement hydration product; besides, a higher addition amount of coalescing agent resulted in the most obvious cross-linking effect; when coalescing agent was not added, and C-S-H was scattered onto the polymer film (refer to SECC). When the coalescing agent addition amount was increased from 0% to 2%, the microstructure of test specimen gradually translated into the cross-linking between C-S-H and the polymer film, and the cross-linking degree was aggravating; at the addition amount of 3%, it developed into the structural morphology of little scattered C-S-H and the approximately integral transition interface; when the addition amount was increased to 4%, it formed the continuous and integral structural morphology with no obvious transition interface (refer to FFAMC-4). Further, it was observed based on Figure 14b–d that, the coalescing agent promoted the polymer film formation, with the increase in its addition amount, on the one hand, the organic polymer film inside the test specimen kept on cross-linking with C-S-H, which gradually formed the continuous and integral transition interface. On the other hand, the polymer film crosses the pore to form the continuous and tight spatial network structure. The polymer film possessed good property in bonding and elasticity, the continuous and integral transition interface enhanced the test specimen strength, the tight spatial network structure boosted the deformation capacities of test specimens under tensile and shear loads, and the external energy was absorbed and translated into the own deformation energy, thus improving the energy consumption property. Consequently, coalescing markedly improved the bond properties, tensile properties, and shear performances of the SECC test specimens, and the FFAMC test specimen achieved the optimal comprehensive mechanical properties at its addition amount of 4%.

It was shown in Figure 14a that, when the nanometer aluminium oxide addition amount was 0.5%, the microstructure of NAMC test specimen showed no obvious change compared with that of SECC test specimen; at the addition amount of 1%, the scattered inorganic powders and C-S-H were reduced; when the addition amount was increased to 1.5%, the scattered inorganic powders and C-S-H were further reduced, an a continuous and smooth structure was formed, as shown by NAMC-3. However, with the further increase in nanometer aluminium oxide addition amount, the nanometer aluminium oxide particles hardly dispersed evenly in the composite material, and nanometer aluminium oxide particle enrichment and aggregation phenomena began to appear, as shown by NAMC-4 and NAMC-5. Further, based on Figure 14e–g, the nanometer aluminium oxide particles had strong absorption capacity and high catalytic activity, when incorporated into SECC, the large amount of calcium hydroxide harmful crystals in the cement hydration products were adsorbed onto the nanometer aluminium oxide surface, which were reacted and consumed to produce the hydrated calcium aluminate; at the same time, the nanometer aluminium oxide particles bonded more nanometer hydration products based on the original cement hardened pastes to develop secondary hydration reaction, thus forming the smooth and continuous film structure, which reduced the enrichment and oriented arrangement of calcium hydroxide harmful crystals, increased the content of hydration product C-S-H on the transition interface, and thereby improved the interface properties of composite materials. Therefore, the bond property, tensile property, and shear performances of the test specimens were improved. Meanwhile, the nanometer aluminium oxide particle enrichment and aggregation phenomena were observed when the nanometer aluminium oxide addition amount was 1.5%, the polymer film was wrapped or absorbed onto the film surface, which prevented it from sufficient reaction, leading to the formation of the weak region in the test specimen. Consequently, the mechanical properties of test specimens were reduced, and the optimal nanometer aluminium oxide addition amount was 1.5%.

### 4.2. Pore Structure Analysis 

Table 7 shows the pore structural parameters of the SECC test specimens modified by different admixtures measured through mercury injection test, Figure 15 presents the effects of different modification manners on the pore distribution differential curve of the composite materials, and Figure 16 illustrates the effects of different modification manners on the pore volumes and percentages of test specimens. Clearly, after coalescing agent was added, the pore volumes and average pore sizes of test specimens were apparently reduced; compared with SECC test specimen, those in FFAMC-4 test specimen were reduced by 49.2% and 40.5%, respectively, the maximum aperture was markedly reduced, the aperture distribution was not obviously changed, and the large aperture percentage was slightly decreased. The possible reasons were that, on the one hand, the coalescing agent cross-linked polymer film molecules with inorganic powders through the chemical bond, which improved the interface bond properties of the two and improved the test specimen compactness; on the other hand, the inorganic powder surface had higher dispersity after the cross-linking with coalescing agent, which reduced the bubble-produced pores introduced due to uneven dispersion. After adding plasticizer, the percentages of medium aperture and large aperture in the test specimens were apparently reduced, the aperture distribution shifted to the reduction direction, but the average aperture and pore volume were evidently increased. Compared with the SECC test specimen, the average aperture and pore volume of PLMC-4 test specimen were increased by 27.2% and 38.3%, respectively. After adding silane coupling agent, the maximum apertures of test specimens were not markedly changed, while the medium aperture was apparently reduced, the aperture direction shifted to the reduction direction, the large pore percentage was decreased, the transition pore and capillary pore percentages were slightly improved. However, the pore volume and average aperture were markedly enhanced, compared with the SECC test specimen, those of SCAMC-4 test specimen were increased by 28.1% and 22.9%, respectively. Thus, it was clear that, plasticizer and silane coupling agent improved the aperture distribution of composite material test specimens, reduced the number of large pore, substantially increased the pore volume and pore size, therefore, their respective addition amounts must be controlled within the reasonable ranges. The major causes of this phenomenon was that, plasticizer and silane coupling agent weakened the intermolecular force between the polymer chains and softened the polymer emulsion particles, respectively, when they were added into the composite material, they promoted the plastic flow of polymer particles, allowed for the sufficient contact between emulsion particles and inorganic powders, and accelerated the film formation, thus improving the aperture distribution of the test specimen. However, when large amounts of plasticizer and silane coupling agent were added, they remarkably weakened or softened the polymer emulsion particles, which increased the intermolecular gap; besides, the plasticizer and silane coupling agent gradually evaporated in the test specimen after they exerted their action, leading to the apparently increased pore volume and average aperture of the test specimen.

After adding nanometer aluminium oxide, the pore volume of NAMC test specimen was remarkably reduced compared with that of SECC test specimen, the characteristic aperture was markedly reduced, the aperture distribution kept on shifting towards the decrease direction, the large pore percentage was apparently reduced, and the gel pore and transition pore percentages were improved. At the nanometer aluminium oxide addition amount of 1.5%, the test specimen had the optimal pore structure, with the pore volume of 0.0716 mL/g, the average pore size of 35.17 nm, the large pore percentage of 57.68%, which were reduced by 48.89%, 65.90%, and 21.43%, respectively, compared with the SECC test specimen. When the nanometer aluminium oxide addition amount was increased to 2.5%, the pore volume, characteristic aperture and large pore percentage of NAMC-5 test specimen were slightly increased relative to those of NAMC-3 test specimen, but obviously decreased compared with those of SECC test specimen. Thus, it was obvious that the addition of appropriate amount of nanometer aluminium oxide effectively reduced the pore volume of test specimen, refined the aperture size, improve aperture distribution, and thus enhanced the test specimen compactness. Nonetheless, excessive nanometer aluminium oxide reduced the modification effect, which was ascribed to the surface effect of nanometer aluminium oxide particles. To be specific, nanometer particle has great specific surface area and high surface energy, which is able to tightly adsorb the inorganic powders and polymer emulsion particles, and to develop chemical bonding with the cement hydration products to form the compact network structure, thus enhancing the test specimen compactness. Moreover, a large amount of water is required for the surface wetting and depolymerization of nanometer particle, which further decreases the pore produced due to water evaporation. However, excessive nanometer aluminium oxide particles adsorb and aggregate together, leading to the increased pores within the test specimen.

To sum up, the addition of coalescing agent, plasticizer, silane coupling agent and nanometer aluminium oxide refined the pores in the SECCs, among which, coalescing agent and nanometer aluminium oxide had the best modification effects, which refined the pore structure, reduced the porosity, increased the compactness of composite material, and thus markedly enhanced the comprehensive mechanical properties of test specimen.

## 5. Conclusions 

Adding appropriate amounts of coalescing agent, plasticizer, silane coupling agent, and nanometer aluminium oxide enhances the bond properties of SECC test specimens. Typically, the optimal bond properties of FFAMC, PLMC, SCAMC, and NAMC test specimens were attained at the coalescing agent, plasticizer, silane coupling agent, and nanometer aluminium oxide addition amounts of 4%, 3%, 1% and 1.5%, respectively.

The improving effects of various admixtures on the tensile properties of SECC test specimens follow the order of coalescing agent > nanometer aluminium oxide > plasticizer > silane coupling agent, and the optimal addition amounts are 4%, 1.5%, 1% and 2%, respectively. Adding appropriate amounts of coalescing agent and nanometer aluminium oxide enhances the tensile strengths, deformation and energy consumption properties of the test specimens; meanwhile, adding appropriate amount of plasticizer increases the tensile strains and energy consumption properties of test specimens, while adding appropriate amount of silane coupling agent only enhances the tensile strengths of test specimens.

The improving effects of various admixtures on the shear properties of SECC test specimens follow the order of coalescing agent > nanometer aluminium oxide > silane coupling agent > plasticizer, and the optimal addition amounts are 4%, 1.5%, 1% and 1%, respectively. Adding appropriate amounts of coalescing agent and nanometer aluminium oxide enhances properties of the test specimens in shear strengths, deformation, and energy consumption; meanwhile, adding appropriate amount of silane coupling agent only enhances the shear strengths of test specimens, whereas adding appropriate amount of plasticizer decreases the shear strength, deformation, and energy consumption properties of test specimens.

The mechanisms of coalescing agent and nanometer aluminium oxide in improving the mechanical properties of SECC test specimens are explained at microscopic level based on the FE-SEM test and MIP test. Of them, coalescing agent enhances the cross-linking between polymer film and cement hydration products inside the test specimen to form the continuous and intact transition interface, reduce the pore volume, and thereby boost the comprehensive mechanical properties of the test specimen. Nanometer aluminium oxide adsorbs the cement hydration products and develops the secondary hydration reaction with them to form the continuous and integral film structure; additionally, it refines the pore structure, reduces the porosity, and enhances the comprehensive mechanical properties of the test specimens.

This work provides an idea for studying high-performance SECC. However, the test in this paper also has limitations. For instance, there are not many test groups based on the amount of different modified admixtures, and the effects of mixing different modified admixtures on the mechanical properties of SECCs need further study.

## Figures and Tables

**Figure 1 materials-13-00008-f001:**
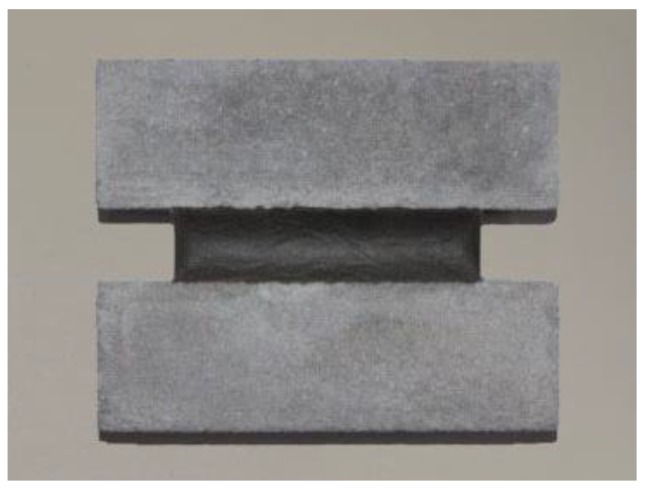
Composite material test specimen.

**Figure 2 materials-13-00008-f002:**
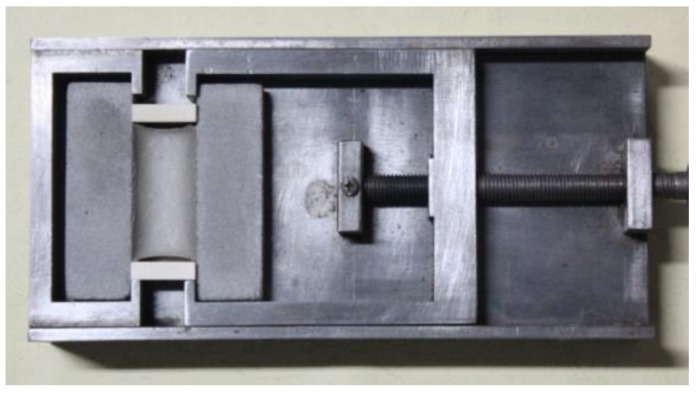
Tensile at definite elongation test mold.

**Figure 3 materials-13-00008-f003:**
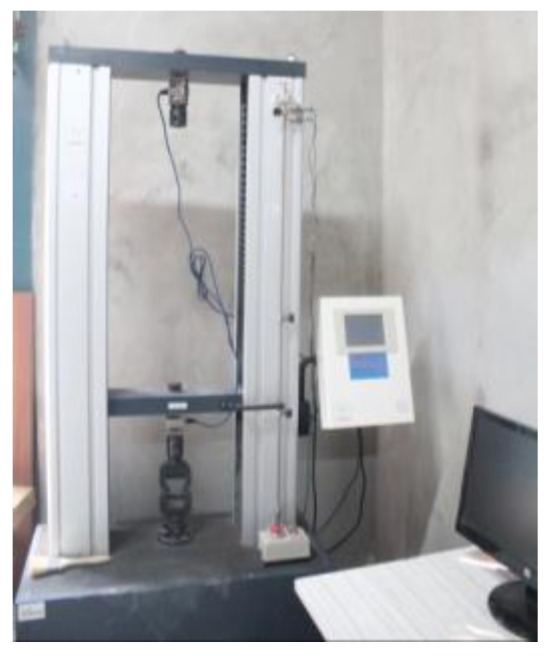
HS-3001B electronic device.

**Figure 4 materials-13-00008-f004:**
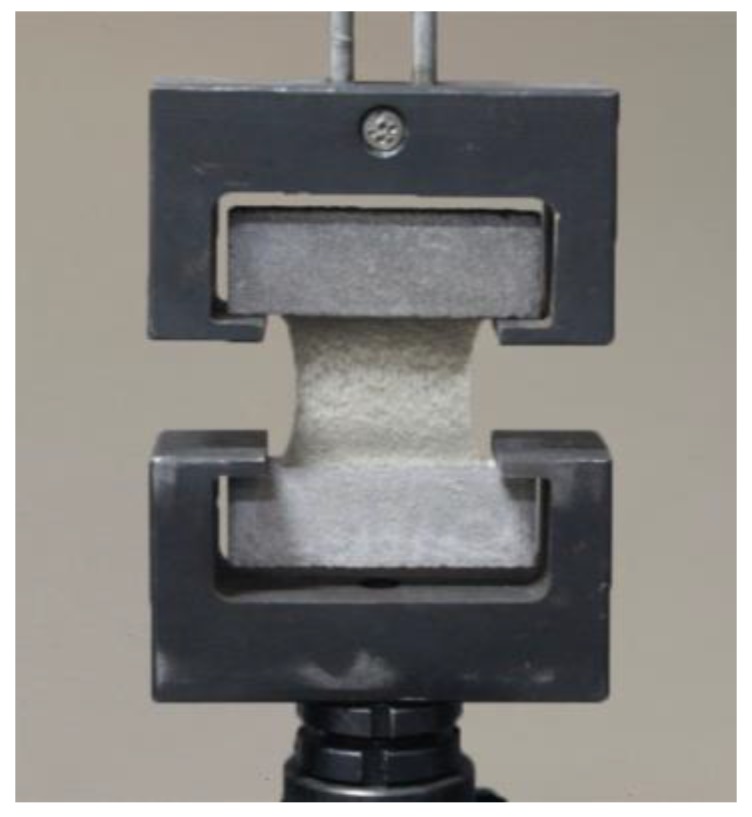
Fixture of tensile at elongation test device.

**Figure 5 materials-13-00008-f005:**
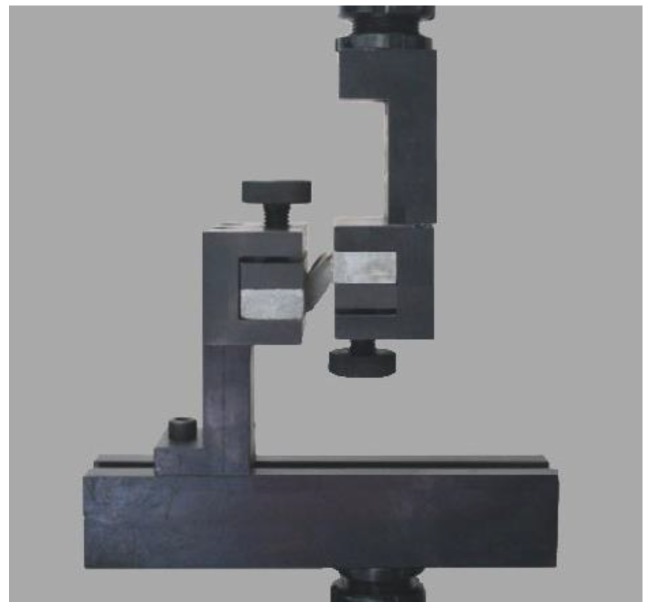
Fixture of shear test device.

**Figure 6 materials-13-00008-f006:**
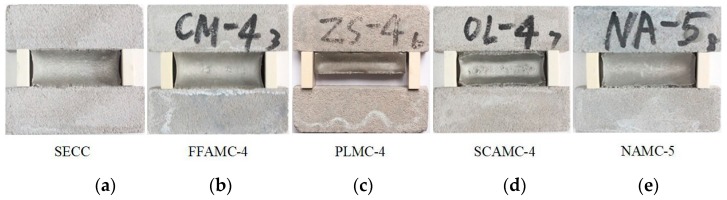
Tension at definite elongation bond failure morphology. (**a**): SECC (**b**): FFAMC-4 (**c**): PLMC-4 (**d**): SCAMC-4 (**e**): NAMC-5.

**Figure 7 materials-13-00008-f007:**
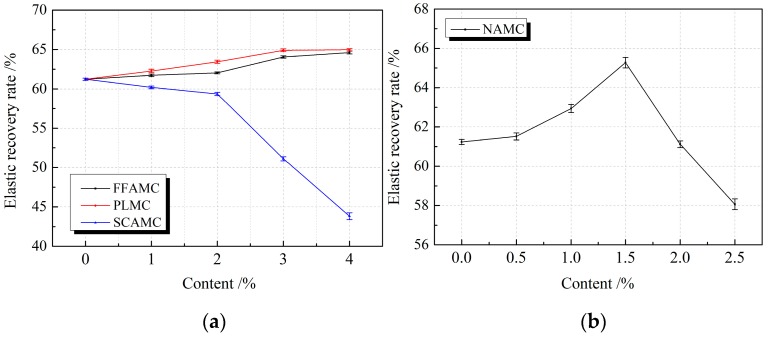
Effects of different admixtures on the elasticity recovery rates of the test specimens. (**a**): Effects of different additives on the elasticity recovery rates of the test specimens (**b**): Effects of nanometer aluminium oxide on the elasticity recovery rates of the test specimens.

**Figure 8 materials-13-00008-f008:**
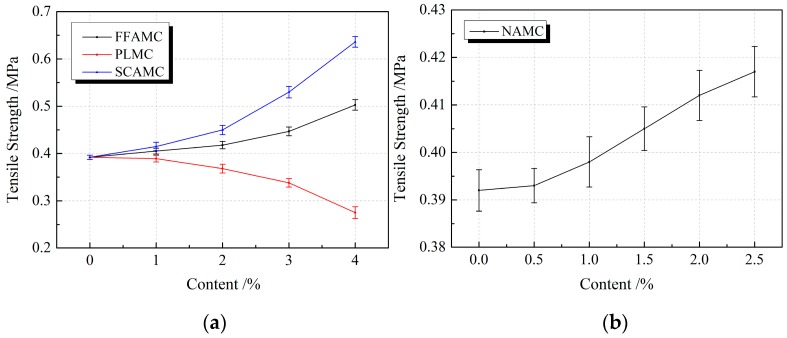
Effects of different admixtures on the tensile strengths of test specimens. (**a**): Effects of different additives on the tensile strengths of test specimens (**b**): Effects of nanometer aluminium oxide on the tensile strengths of test specimens.

**Figure 9 materials-13-00008-f009:**
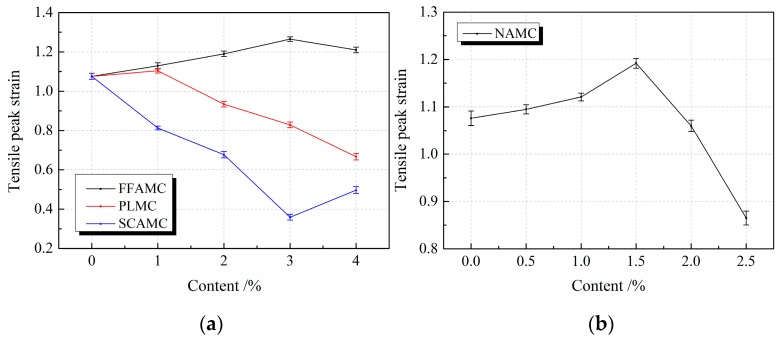
Effects of different admixtures on the tensile peak strains of test specimens. (**a**): Effects of different additives on the tensile peak strains of test specimens (**b**): Effects of nanometer aluminium oxide on the tensile peak strains of test specimens.

**Figure 10 materials-13-00008-f010:**
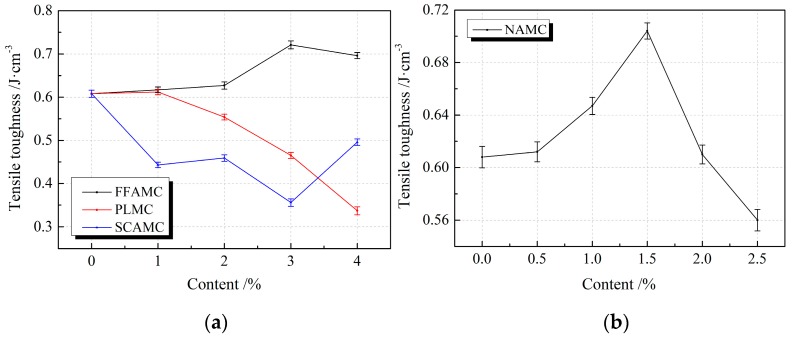
Effects of different admixtures on the tensile toughness of test specimens. (**a**): Effects of different additives on the tensile toughness of test specimens (**b**): Effects of nanometer aluminium oxide on the tensile toughness of test specimens.

**Figure 11 materials-13-00008-f011:**
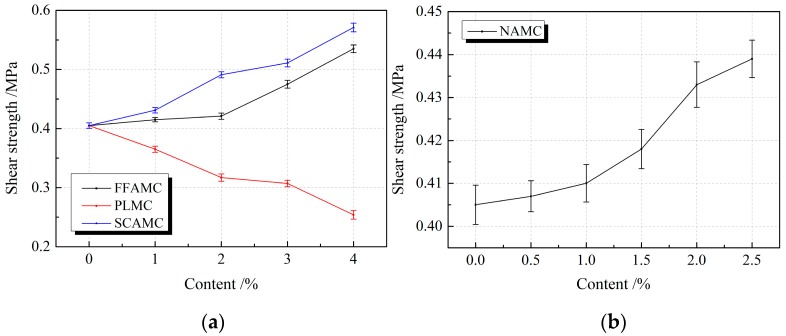
Effects of different admixtures on the shear strengths of test specimens. (**a**): Effects of different additives on the shear strengths of test specimens (**b**): Effects of nanometer aluminium oxide on the shear strengths of test specimens.

**Figure 12 materials-13-00008-f012:**
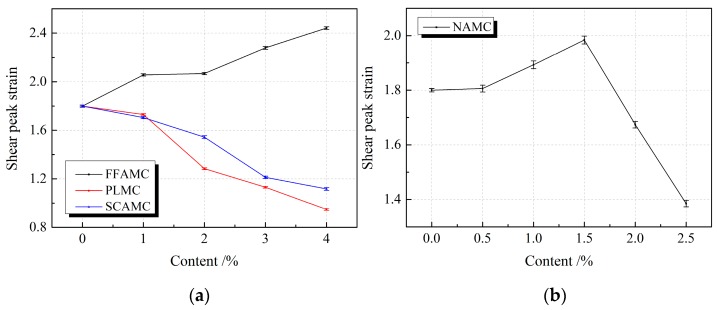
Effects of different admixtures on the shear peak strains of test specimens. (**a**): Effects of different additives on the shear peak strains of test specimens (**b**): Effects of nanometer aluminium oxide on the shear peak strains of test specimens.

**Figure 13 materials-13-00008-f013:**
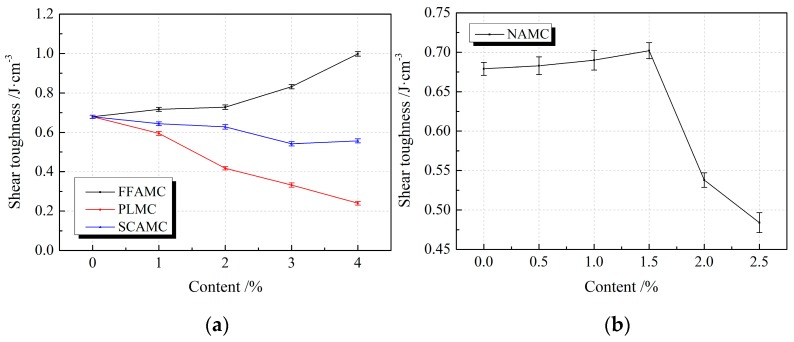
Effects of different admixtures on the shear toughness of test specimens. (**a**): Effects of different additives on the shear toughness of test specimens (**b**): Effects of nanometer aluminium oxide on the shear toughness of test specimens.

**Figure 14 materials-13-00008-f014:**
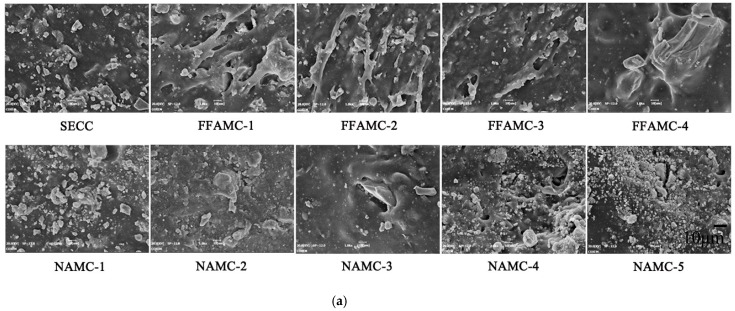
Microstructures of composite material test specimens. (**a**): Microstructures of SECC, FFAMC and NAMC test specimens (**b**): C-S-H (**c**): Network Structure (**d**): Pores (**e**): Polymer Film (**f**): Inorganic Component (**g**): Nano Alumina oxide Particles.

**Figure 15 materials-13-00008-f015:**
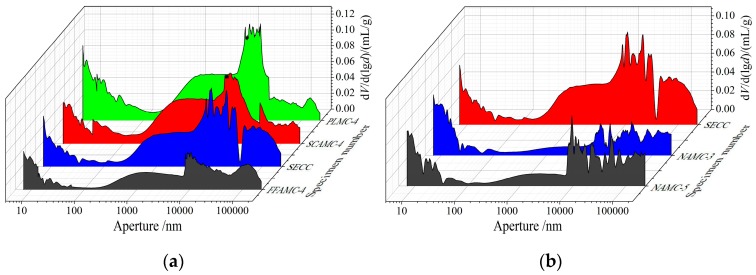
Effects of different modifying admixtures on the aperture distribution differential curves of composite material test specimens. (**a**): Effects of different modifying additives on the aperture distribution differential curves of composite material test specimens (**b**): Effects of nanometer aluminium oxide on the aperture distribution differential curves of composite material test specimens.

**Figure 16 materials-13-00008-f016:**
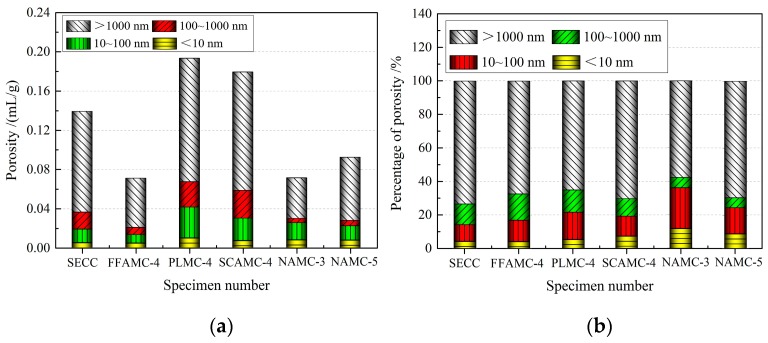
Effects of different modifying admixtures on the pore volumes and percentages of composite material test specimens. (**a**): Effects of different modifying additives on the pore volumes and percentages of composite material test specimens (**b**): Effects of nanometer aluminium oxide on the pore volumes and percentages of composite material test specimens.

**Table 1 materials-13-00008-t001:** Performance indexes of Acronal S400F ap-type styrene-acrylic emulsion.

Appearance	Average Particle Size /μm	Glass Transition Temperature/°C	Solid Content /%	Viscosity /mPa·s	pH Value
Milky white liquid	0.1	-6	55~57	400~1800	7.0~8.4

**Table 2 materials-13-00008-t002:** Performance indexes of DN-12.

Appearance	Molecular Formula	Content	Moisture	Acidity
Colorless, Transparent, No Mechanical Impurity	C_12_H_24_O_3_	≥99.0%	≤0.1%	≤0.05%

**Table 3 materials-13-00008-t003:** Major technical indexes of plasticizer.

Molecular Weight	Melting Point/°C	Boiling Point/°C	Purity/%	Density /g/mL	Water Solubility
390.55	−50	386	99	0.985	Insoluble

**Table 4 materials-13-00008-t004:** Major technical indexes of silane coupling agent.

Specific Gravity	Boiling Point /°C	Refractive Index	Content /%	Water Solubility	Appearance
1.0210	150	1.4875	98	Insoluble	Yellowish butter-like

**Table 5 materials-13-00008-t005:** Major technical indexes of nanometer aluminium oxide.

Crystal Form	Average Particle Size /μm	Specific Surface Area /m^2^/g	Bulk Density /g/cm^3^/	Content /%	Appearance
α	0.2~3	6~12	0.4~0.6	99.99	White powder

**Table 6 materials-13-00008-t006:** Mixture ratios of the composite materials.

Test Specimen Number.	Powder Liquid Ratio	Ash Powder Ratio	Addition Amount of Coalescing Agent	Addition Amount of Plasticizer	Addition Amount of Silane Coupling Agent	Addition Amount of Nanometer Aluminium Oxide	Styrene-Acrylic Emulsion/g	Cement/g	Talcum Powder /g	Dispersing Agent/g	Defoamer /g
SECC	0.40	25%	0	0	0	0	100	10	30	1.12	0.7
FFAMC-1	0.40	25%	1%	0	0	0	100	10	30	1.12	0.7
FFAMC-2	0.40	25%	2%	0	0	0	100	10	30	1.12	0.7
FFAMC-3	0.40	25%	3%	0	0	0	100	10	30	1.12	0.7
FFAMC-4	0.40	25%	4%	0	0	0	100	10	30	1.12	0.7
PLMC-1	0.40	25%	0	1%	0	0	100	10	30	1.12	0.7
PLMC-2	0.40	25%	0	2%	0	0	100	10	30	1.12	0.7
PLMC-3	0.40	25%	0	3%	0	0	100	10	30	1.12	0.7
PLMC-4	0.40	25%	0	4%	0	0	100	10	30	1.12	0.7
SCAMC-1	0.40	25%	0	0	1%	0	100	10	30	1.12	0.7
SCAMC-2	0.40	25%	0	0	2%	0	100	10	30	1.12	0.7
SCAMC-3	0.40	25%	0	0	3%	0	100	10	30	1.12	0.7
SCAMC-4	0.40	25%	0	0	4%	0	100	10	30	1.12	0.7
NAMC-1	0.40	25%	0	0	0	0.5%	100	10	30	1.12	0.7
NAMC-2	0.40	25%	0	0	0	1%	100	10	30	1.12	0.7
NAMC-3	0.40	25%	0	0	0	1.5%	100	10	30	1.12	0.7
NAMC-4	0.40	25%	0	0	0	2%	100	10	30	1.12	0.7
NAMC-5	0.40	25%	0	0	0	2.5%	100	10	30	1.12	0.7

Notes: (1) the addition amounts of coalescing agent, plasticizer, silane coupling agent, and nanometer aluminium oxide were the respective mass percentages of the styrene-acrylic emulsion mass; (2) the addition amounts of dispersing agent and defoamer were 0.8% and 0.5% of the total masses of styrene-acrylic emulsion and the inorganic powders, respectively.

**Table 7 materials-13-00008-t007:** Pore structural parameters of composite material test specimens.

Specimen Number	Total Pore Volume /mL/g	Average Aperture /nm	Median Aperture /nm	Optimum Aperture /nm	Percentage of Porosity/%			
					<10 nm	10~100 nm	100~1000 nm	>1000 nm
SECC	0.1401	103.15	9067.39	10,955.03	4.43	9.80	12.35	73.42
FFAMC-4	0.0712	61.39	7692.27	7.15	7.42	11.84	10.42	70.31
PLMC-4	0.1937	131.16	2971.44	16,718.36	5.39	16.18	13.33	65.09
SCAMC-4	0.1795	126.73	2995.84	10,519.03	4.26	12.67	15.70	67.36
NAMC-3	0.0716	35.17	4198.00	7.16	11.96	24.20	6.16	57.68
NAMC-5	0.0926	51.11	10,030.00	9575.31	8.96	15.57	5.86	69.61

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
