# Peer review of "Comparative Study on the Effects of Various Modified Admixtures on the Mechanical Properties of Styrene-Acrylic Emulsion-Based Cement Composite Materials"

_materials, 2019, doi:10.3390/ma13010008_

Round 1

Reviewer 1 Report

The paper "Comparative Study on the Effects of Various Modified Admixtures on the Mechanical Properties of Styrene-acrylic Emulsion-based Cement Composite Materials" falls within the scope of Materials Journal and shows technical relevance.

In this paper the authors present an interesting study comparing  several admixture-modified styrene-acrylic emulsion-based cement composites(SECCs), and measured the strength, deformation, and energy consumption indexes of test  specimens, so as to investigate the influences of coalescing agent, plasticizer, silane coupling agent, and nanometer aluminium oxide on  the bond, tensile and shear mechanical properties of the test specimens.

The material is publishable, but requires improvement. In this sense, there are some suggestions on the attached paper that should be addressed before publishing.

Suggestion 01

Novelty unclear: What is the original contribution of the study? The introduction section is not very enlightening on the subject. Novelty should be made as clear as possible.

Suggestion 02

The limitations of the study should also be included.

Suggestion 03

Some units are missed in Table 6.

Suggestion 04

The results of unmodified SECC specimens are not shown in graphs presented in Figures 7, 8, 9, 10, 11, 12 and 13. These data would be useful for understanding the real scope of the paper.

Suggestion 05

In section 4.1 the method for determining the microstructure is not specified.

Suggestion 06

In Figure 14 a) to g) it is very difficult to identify the (SEM?) pictures magnification degree.

Author Response

Dear Editors and Reviewers,

We deeply appreciate the time and effort you have spent in reviewing our manuscript entitled “Comparative Study on the Effects of Various Modified Admixtures on the Mechanical Properties of Styrene-acrylic Emulsion-based Cement Composite Materials(Manuscript ID: materials-654942). Your comments are all valuable and very helpful for revising and improving our paper. We have studied the comments carefully, and have made revisions point-by-point. All changes made to the previous manuscript are marked in the revised manuscript. The main corrections in the paper and the responds to the reviewers’ comments are as follows.

Comment 1: Novelty unclear: What is the original contribution of the study? The introduction section is not very enlightening on the subject. Novelty should be made as clear as possible.

Response: Many thanks for the reviewer’s suggestions. This suggestion is of great significance to the scientificity of this paper. As reviewer suggested, we have revised the Introduction section to emphasize the novelty of the work. We have added analysis of the status and limitations of current research at home and abroad, and clarified the value of our work. More details can be seen in the revised manuscript. Thanks again for the reviewer addressing this issue.

Comment 2: The limitations of the study should also be included.

Response: We thank the reviewer for addressing this issue. We are sorry that the original description lacks the limitations of the study. We have added Section 5 in the revised manuscript, where the work in this paper and the limitations of the study have been analyzed in detail. More details can be seen in Section 5 of the revised manuscript. Special thanks to you for your good comments.

Comment 3: Some units are missed in Table 6.

Response: Thanks for the reviewer’s suggestion. We are sorry for our negligence of this point. We have added necessary units in Table 6 in the revised manuscript. Special thanks to you for your good comments.

Comment 4: The results of unmodified SECC specimens are not shown in graphs presented in Figures 7, 8, 9, 10, 11, 12 and 13. These data would be useful for understanding the real scope of the paper.

Response: Thanks for the reviewer’s suggestion. As reviewer mentioned, the results of unmodified SECC specimens are useful for understanding the real scope of the paper. We are sorry that we did not make it clear. In fact, these data are shown in graphs presented in Figures 7, 8, 9, 10, 11, 12 and 13, which are the points in graphs presented in these Figures when the content of different admixtures is 0. More details can be seen in the Figures 7, 8, 9, 10, 11, 12 and 13 of the revised manuscript. Special thanks to you for your good comments.

Comment 5: In section 4.1 the method for determining the microstructure is not specified.

Response: Thanks for the reviewer’s suggestion. We are sorry that we did not make it clear. We have introduced the method for determining the microstructure in Section 2.3, where we have described the method of FE-SEM test and MIP test in detail. More details can be seen in the revised manuscript. Special thanks to you for your good comments.

Comment 6: In Figure 14 a) to g) it is very difficult to identify the (SEM?) pictures magnification degree.

Response: Thanks for the reviewer’s suggestion. We are sorry for our negligence of this point. As you suggested, we have indicated the pictures magnification degree in Figure 14 (a) to (g) of the revised manuscript. Specially, the magnification degree of all pictures in the Figure 14 (a) are same. More details can be seen in Section 4.1 in the revised manuscript. Thanks again for the reviewer addressing this issue.

Once again, we appreciate for Editors/Reviewers’ warm work earnestly, and hope that the revision will meet with approval.

Thank you and best regards.

Sincerely yours,

Tengjiao Wang

December 3, 2019

Reviewer 2 Report

The manuscript presents a very interesting topic and concerns the  evaluation of various  modified admixtures on  the mechanical properties of cement.  The subject matter is within the scope of the journal. The methodology is sufficiently well explained that someone else knowledgeable about the field could repeat the study.  Each figure and table is necessary to the understanding of the conclusions. All elements of the manuscript relate logically to the study's statement of purpose.  The work is well written but needs some adjustments.  As a conclusion of this manuscript is acceptable for publication after the minor revision. Some suggestions for improvement are given as follows:

Abstract:
Authors underscore the scientific value added the in the abstract. The indication of some of the most critical quantitative results to the Abstract, contributed to better understanding and readability. To sum up, the Abstract  does not require any corrections.

Keywords:
Consider using shorter Keywords, e.g. instead of "styrene acrylic emulsion based styrene acrylic emulsion" you can use "styrene acrylic emulsion" and "cement composite".

Introduction:
Introduction requires minor amendments:
- Damaged pavement airport  is a threat mainly when taking off and landing an aircraft.   Describe "hidden dangers to the safety flight of aircraft".

Tests:
|The description of the research methods is very detailed. However, considering the non-standard way of preparing sample, the point "Tests" should be considered very good.

Test results and analysis:
Are the values shown in the drawings average values? Complete the descriptions with the number of samples tested. Also add statistical analysis of the results.

Micro-analysis:
Excellent discussion of results.

Conclusion:
The conclusions relate to conducted research, they do not require improvement.

Author Response

Dear Editors and Reviewers,

We deeply appreciate the time and effort you have spent in reviewing our manuscript entitled “Comparative Study on the Effects of Various Modified Admixtures on the Mechanical Properties of Styrene-acrylic Emulsion-based Cement Composite Materials(Manuscript ID: materials-654942). Your comments are all valuable and very helpful for revising and improving our paper. We have studied the comments carefully, and have made revisions point-by-point. All changes made to the previous manuscript are marked in the revised manuscript. The main corrections in the paper and the responds to the reviewers’ comments are as follows.

Comment 1: Consider using shorter Keywords, e.g. instead of "styrene acrylic emulsion based styrene acrylic emulsion" you can use "styrene acrylic emulsion" and "cement composite".

Response: Many thanks for the reviewer’s suggestions. As reviewer suggested, we have revised the long keyword. More details can be seen in the revised manuscript. Special thanks to you for your good comments.

Comment 2: Introduction requires minor amendments:

-Damaged pavement airport is a threat mainly when taking off and landing an aircraft.  Describe "hidden dangers to the safety flight of aircraft".

Response: Thanks for the reviewer’s suggestion. As you mentioned, we have revised the sentence in Section 1. More details can be seen in the revised manuscript. Thanks again for the reviewer addressing this issue.

Comment 3: Test results and analysis:

Are the values shown in the drawings average values? Complete the descriptions with the number of samples tested. Also add statistical analysis of the results.

Response: Many thanks for the reviewer’s suggestions. Yes, all the values shown in the drawings are average values. Each value represents the average value from three independent test results of samples. As reviewer suggested, we have added error bars in the graphs to obtain statistical analysis of the results. More details can be seen in the Figures of the revised manuscript. Special thanks to you for your good comments.

Once again, we appreciate for Editors/Reviewers’ warm work earnestly, and hope that the revision will meet with approval.

Thank you and best regards.

Sincerely yours,

Tengjiao Wang

December 3, 2019

Round 2

Reviewer 1 Report

The revised manuscript includes all the recommendations of this reviewer, so it is accepted in its current form.

Author Response

Dear Reviewer,

We deeply appreciate the time and effort you have spent in reviewing our manuscript. It's pleased for us to receive your new comments. Thank you very much.

Best regards.

Sincerely yours,

Tengjiao Wang

December 10, 2019